

# Development of the MPAS-CMAQ Coupled System (V1.0) for Multiscale Global Air Quality Modeling

David C. Wong[1,*], Jeff Willison[1,*], Jonathan E. Pleim[1], Golam Sarwar[1], James Beidler[1], Russ Bullock[1], Jerold A. Herwehe[1], Rob Gilliam[1], Daiwen Kang[1], Christian Hogrefe[1], George Pouliot[1], and Hosein Foroutan[2]

[1]Office of Research and Development, U.S. Environmental Protection Agency, Durham, NC USA
[2]Civil & Environmental Engineering, Virginia Tech, Blacksburg, VA USA
[*]These authors contributed equally to this work.

**Correspondence:** David C. Wong (wong.david-c@epa.gov) Jeff Willison (willison.jeff@epa.gov) Jon Pleim (pleim.jon@epa.gov)

**Abstract.**

The Community Multiscale Air Quality (CMAQ) model has been used for regulatory purposes at the US EPA and in the research community for decades. In 2012, we released the WRF-CMAQ coupled model that enables aerosol information from CMAQ to affect meteorological processes through direct effects on shortwave radiation. Both CMAQ and WRF-CMAQ are considered limited area models. Recently, we have extended domain coverage to global scale linking the meteorological Model for Prediction Across Scales - Atmosphere (MPAS-A, hereafter referred simply to as MPAS) with CMAQ to form the MPAS-CMAQ global coupled model. To configure these three different models, i.e. CMAQ (offline), WRF-CMAQ, and MPAS-CMAQ, we have developed the Advanced Air Quality Modelling System (AAQMS) for constructing each of them effortlessly. We evaluate this newly-built MPAS-CMAQ coupled model using two global configurations: a 120 km uniform mesh and a 92-25 km variable mesh with the finer area over North America. Preliminary computational tests show good scalability and model evaluation, a three years simulation (2014 - 2016) for the uniform mesh case and a monthly simulation of January and July 2016 for the variable mesh case, on ozone and $PM_{2.5}$, show reasonable performance with respect to observations. The 92-25 km configuration has a high bias in wintertime surface ozone across the United States and this bias is consistent with the 120 km result. Summertime surface ozone in the 92-25 km configuration is less biased than the 120 km case. The MPAS-CMAQ system reasonably reproduces the daily variability of daily average PM from the Air Quality System (AQS) network.

## 1 Introduction

The Community Multiscale Air Quality (CMAQ) model (Byun and Schere, 2006) was developed at the U.S. Environmental Protection Agency (EPA) starting in the mid-1990s. CMAQ is a Eulerian 3-D chemistry transport model currently being used both as a regulatory model at the EPA and a research tool by scientists around the world to study various air pollution problems on regional to hemispheric scales. In offline mode, CMAQ is driven by 3-D meteorological fields provided by upstream simulations using meteorology models, such as the Weather Research and Forecasting (WRF) model (Skamarock et al., 2008).





The Meteorology-Chemistry Interface Processor (MCIP) (Otte and Pleim, 2010) is used to provide model-ready meteorological data to the CMAQ model. The process includes unit conversions, format conversions, and vertical grid resolution-related interpolations, as well as calculating additional diagnostic variables required by CMAQ not available in the meteorology model output. In addition to the hourly meteorological data typically produced by MCIP, running CMAQ in offline mode also relies on initial and lateral boundary conditions for chemical constituents (which are created by extraction and interpolation from a larger scale chemistry-transport model), and emissions data. Drawbacks of the regional-to-hemispheric offline configuration are errors introduced by the constant temporal interpolation of meteorological data to a advection time step needed by CMAQ, as well as a lack of feedbacks between the air quality model and the meteorological model (note that cmaq impacts WRf and in turn WRF impacts cmaq and thus it is a two-way interaction).

These drawbacks led to linking CMAQ with WRF to form the WRF-CMAQ coupled model in 2008 (Wong et al., 2012) which included aerosol direct radiative effects. This is an online model where the meteorological information is sent to CMAQ at a user-specified frequency and the aerosol information is fed back to WRF, affecting the short-wave radiation calculation. The implementation of aerosol radiative effects in the WRF-CMAQ model and a demonstration of its impacts on simulated meteorology and subsequent air quality (Wong et al., 2012) led to interest in exploring such interactions in heavily polluted environments (Wang et al., September 2014; Xing et al., 2015), and wildfire scenarios (Wong et al., 2016). The coupled online system also eliminates meteorological temporal interpolation errors that affect offline (sequential) meteorology-AQ systems. More recently, the CMAQ model and the WRF-CMAQ coupled model were extended from regional scale to hemispheric coverage (Mathur et al., 2017).

However, there are still limitations to the WRF-CMAQ coupled model, which include inconsistencies in dynamics between WRF and CMAQ, boundary condition requirements from different global chemistry models which compound into a chemical species mapping problem, and grid structure differences between global and regional models within the WRF-CMAQ coupled model. In addition, multiple grid nesting from hemispheric to local scales adds interpolation errors at every step of refinement. The National Center for Atmospheric Research (NCAR) has recently developed a new global meteorological model, the Model for Prediction Across Scales – Atmosphere (MPAS-A, hereafter referred to as MPAS), which uses a predominantly hexagonal mesh to provide spatial refinement that minimizes discontinuities from global to regional scales. MPAS can be configured with high resolution for regions of interest with seamless transition to coarser resolution for the rest of the globe. In this study, we describe the development of a system to link MPAS with CMAQ to form a global coupled modeling system. This new modeling system will eliminate need for multiple nested modeling domains as well as spatial interpolation errors to enable more robust examinations of the impacts of international transport on modulating background concentrations that will aid multiple entities including the national ambient air quality standards (NAAQS) and continued progress towards regional haze goals.

A brief description of the numerical model components of the MPAS-CMAQ coupled model is provided in Section 2. The depiction of the other components of the MPAS-CMAQ coupled model is given in Section 3. Section 4 presents the general performance of this new coupling system, while Section 5 summarizes the main results and lays out future work.





## 2 Overview of scientific components of the coupled model

The MPAS-CMAQ coupled model was constructed based upon the development of the Advanced Air Quality Modelling System (AAQMS) platform (Fig. 1). This platform provides a flexible environment for modelers to construct the offline CMAQ model, the WRF-CMAQ coupled model, or the MPAS-CMAQ coupled model. The meteorological and air quality model layers of the AAQMS will be described in this section, and the other two layers, unified coupler, and MIO (Model I/O), will be described in section 3.

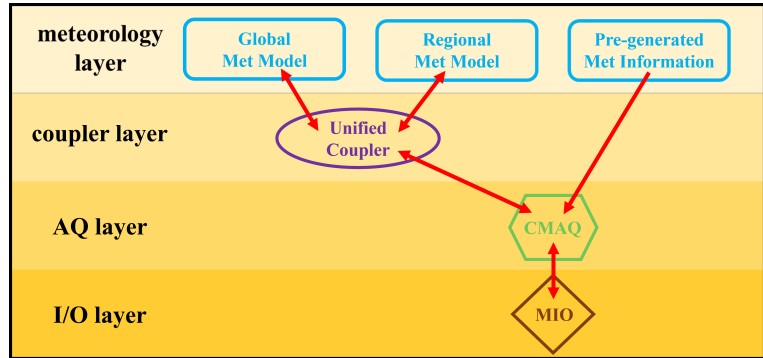

**Figure 1.** Synopsis of the Advanced Air Quality Modeling System (AAQMS). Arrows indicate the flow of information.

### 2.1 MPAS

MPAS is a global meteorological model performing computations on an unstructured mesh with primarily hexagons, as well as some pentagons and heptagons. Unstructured meshes can be uniform (Fig 2 left) or with one or more focus area(s) of seamless mesh refinement (Fig. 2 right). The dynamical equations numerically solved by the MPAS model are fully compressible, Euler nonhydrostatic, and are conservative for all scalar variables. The prognostic variables are the three velocity components, perturbation potential temperature, perturbation geopotential, and perturbation air surface pressure. Additional prognostic variables depend on the model physics options and may include turbulent kinetic energy, water vapor mixing ratio, and several cloud microphysical scalars such as cloud water and ice mixing ratio, rain and snow mixing ratio, and graupel mixing ratio. Both the MPAS and the CMAQ model are configured with the exact same grid configurations and coordinate systems. Thus, no spatial interpolation of either meteorological or chemical data is required.

MPAS source code is publicly available on github (https://github.com/MPAS-Dev/MPAS-Model). The MPAS portion of this coupled model is based on version 7.0 with additional modifications described in the following two sub-sections.

### 2.1.1 MPAS model enhancements

Additional enhancements were necessary in order to use MPAS for retrospective air quality simulations. Specifically, several physics options, namely the Pleim-Xiu land-surface model (PX LSM) (Pleim and Xiu, 1995, 2003; Xiu and Pleim, 2001), the





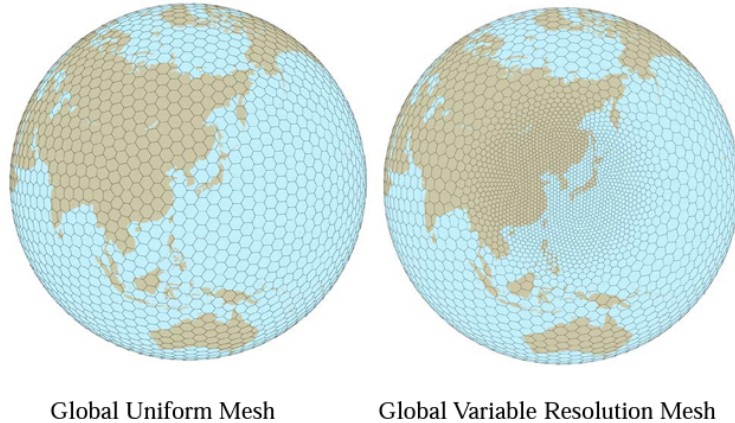

Global Uniform Mesh      Global Variable Resolution Mesh

**Figure 2.** MPAS unstructured uniform mesh and unstructured variable mesh with one area of refinement. source: https://www.ecmwf.int/sites/default/files/elibrary/2012/14043-global-nonhydrostatic-atmospheric-model-mpas-preliminary-results-uniform-and-variable.pdf

Asymmetric Convective Model 2 (ACM2) planetary boundary layer scheme (Pleim, 2007a, b), the Pleim surface layer scheme (Pleim, 2006), and the subgrid-scale convective cloud feedback to the radiation schemes (Alapaty et al., 2012) were added to MPAS, as these options are critical to improve model performance for retrospective air quality applications. Gilliam et al. (2021) provides a thorough MPAS model performance evaluation with these new options. Additionally, the Four-Dimensional

Data Assimilation (FDDA), using a method similar to analysis nudging in WRF but adapted to the polygonal Voronoi mesh in MPAS, was added. The FDDA method is described in (Bullock Jr. et al., 2018) along with test simulations conducted for January and July 2013 showing the new FDDA option constrains model errors relative to both the target fields used for FDDA and standard meteorological observations, while still maintaining the conservation of mass reasonably.

### 2.1.2 Transport processes

The WRF-CMAQ coupled model is designed to have two independent and different transport algorithms within WRF and CMAQ to handle the dynamic processes and chemical species, respectively. However, in the MPAS-CMAQ coupled model, the transport portion of the code within CMAQ is turned off and all transport is handled by the MPAS model for consistency. At the end of the CMAQ step, all the chemical species information is transferred to MPAS as scalars. Once they have gone through the transport process and the end of the MPAS step has been reached, they are then transferred back to CMAQ and

this cycle repeats itself. These new scalars are added in the MPAS Registry.xml file before compilation. Since CMAQ supports multiple chemical mechanisms with different species, a simple Fortran code was created as a tool to facilitate adding new scalars into the registry file.





## 2.2 CMAQ

CMAQ (Appel et al., 2021) is a 3-D Eulerian atmospheric chemistry and transport model that numerically integrates a set of in-
terdependent mass conservation equations for various chemical species. The CMAQ model employs operator splitting to mod-
ularize the various physical and chemical processes including subgrid turbulent vertical transport, horizontal and vertical ad-
vection, horizontal diffusion, cloud processes (i.e., aqueous chemistry, subgrid convective transport, wet deposition), gas-phase
chemistry, and aerosol chemistry and dynamics. The CMAQ system also ingests anthropogenic and wildfire emission rates typ-
ically processed by the Sparse Matrix Operator Kernel Emissions (SMOKE, http://www.cep.unc.edu/empd/products/smoke).
Plume rise, biogenic and natural emissions, and dry deposition are all modeled components of the CMAQ model. Both sources
(emissions) and sinks (deposition) are applied as mass tendencies in the vertical diffusion calculation. In this study we use
CMAQ v5.4, publicly available on github (https://github.com/USEPA/CMAQ).

## 3 Other Layers of the Coupled Model

The details of the remaining layers of the AAQMS that facilitate the construction of the MPAS-CMAQ coupled model are
presented here. The unified coupler works with both the WRF-CMAQ and the MPAS-CMAQ coupled models. The I/O layer
deals with I/O on the CMAQ side only.

### 3.1 Unified Coupler

The Earth System Modeling Framework (ESMF) (Hill et al., 2004) is a popular model coupler, that provides a means of
exchanging data between two models. It also handles map projection conversion when the models are simulating in two
different map projections, and data re-mapping/interpolation when the domain coverage and resolution are not the same in
both models to ensure data exchange is done properly. There are also some couplers that are tailored for a specific application
(Craig et al., 2005; Larson et al., 2005).

In this work, we adopted a simpler approach tailored for our specific application. For both the WRF-CMAQ and the MPAS-
CMAQ coupled models, CMAQ inherits the domain structure from either WRF or MPAS, and therefore, the coupler inherits
the map projection, grid alignment, and grid spacing seamlessly. For simplicity, we just need a straightforward mechanism for
data exchange between the two models (Fig. 3).

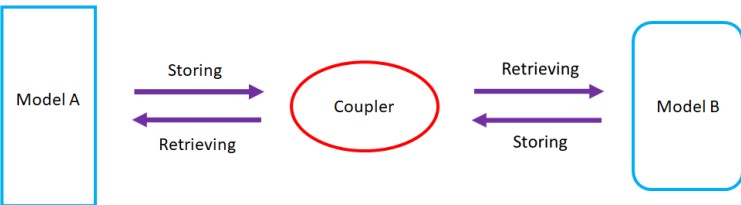

**Figure 3.** Depiction of the Unified Coupler.





The coupler in the WRF-CMAQ coupled model was based on the BUFFERED file (in memory) feature from IOAPI3 (Input/Output Application Programming Interface version 3, https://www.cmascenter.org/ioapi). CMAQ utilizes IOAPI3 to provide basic I/O functions and other supplementary functions such as time to second conversion. Due to various constraints such as character string length is 16, and lack of flexibility such as all variables are with the dimensionality, creation of this unified coupler is the first step of moving away from IOAPI3 while constructing a single universal coupler serving both the WRF-CMAQ and the MPAS-CMAQ coupled models for software interoperability purposes. The data exchange occurs in both directions. Meteorological data is made available to drive the CMAQ model, and subsequently aerosol information is passed back to the meteorological model so that it can affect the radiation calculations (aerosol radiative direct effect). However, we should note that at this stage of the MPAS-CMAQ development, further work is needed to implement a robust and flexible way to represent soluble and insoluble lumped species, which are two of the five categorical components impacting the radiation calculations, with respect to different chemical mechanisms supported by CMAQ. Hence, the current MPAS-CMAQ coupled model does not yet include aerosol radiative direct effect but the coupler infrastructure will allow its implementation in future work.

The unified coupler design addresses two major characteristic differences among the WRF-CMAQ and MPAS-CMAQ coupled models. Firstly, the WRF model decomposes the horizonal domain with rectangular grids and stores information using a 2-D data structure. In contrast, the MPAS model decomposes the horizonal domain with an unstructured mesh and stores information using a 1-D data structure. Secondly, in the WRF-CMAQ coupled model, the WRF time step could be sub-divided by the advection time step algorithm on the CMAQ side. In addition, two time steps of meteorological data are required and stored in a circular buffer to support interpolation. With the MPAS-CMAQ coupled model, the transport is handled in MPAS and the time step in CMAQ is fully synchronized with MPAS.

### 3.1.1 MIO

All input and output files associated with MPAS, WRF, and CMAQ are in the Network Common Data Form (netCDF, http://www.unidata.ucar.edu/packages/netcdf/index.html) (Rew and Davis, 1990). MPAS and WRF have their own input/output (I/O) systems, while CMAQ I/O is handled by IOAPI3. The key difference is the definition and requirement of the data declaration section of the netCDF file among all these three models. The IOAPI3 I/O format has less flexibility, e.g., all variables in a given file must have the same dimensionality. Reading in emission inputs, writing calculated concentration fields, and diagnostic information are the main functions of the I/O system in CMAQ. Running the coupled model, whether it is MPAS-CMAQ or WRF-CMAQ, needs initial conditions (or the restart file), and other auxiliary input files, as well as emission files, which are critical for the chemistry transport model (CTM) CMAQ. We have examined the possibility of having the MPAS I/O system (Dennis et al., 2012) read in emission data. In this case, the MPAS I/O system would read in the emission data, the unified coupler would handle the transfer, and finally, the CMAQ model would ingest this information. For this scenario, the MPAS registry file must be modified to include the emission species information, which is chemical mechanism dependent. In our testing, we have found this method not to be an efficient or practical pathway to reading in emission data.





To streamline I/O processes, we have developed the MIO to provide an interoperable I/O system for reading and writing
in the MPAS-CMAQ, WRF-CMAQ, and CMAQ standalone models in various flavors of netCDF formats. MIO supports the
traditional "pseudo" parallel output paradigm, i.e., each processor sends its own sub-domain data to the I/O processor, typically
PE 0, and the I/O processor stitches the data together and writes it to the disk. MIO can also support true parallel I/O paradigm
utilizing the pnetCDF library and an existing parallel file system (currently not supported for MPAS-CMAQ model). This
feature has been reported in the offline CMAQ case study (Wong et al., 2015).

## 4   Coupled Model Performance

We examine the performance of this newly developed coupled system in two different ways: computational and physical aspects
based on two mesh configurations (120 km uniform mesh and 92-25 km variable mesh). We did a 7-day simulation with the
120 km uniform (1/1/2016 - 1/7/2016) and a one-day (1/1/2016) test with the 92-25 km mesh to determine the computational
performance. Subsequently, we did a monthly test of January and July 2016 to validate the model performance for both mesh
configurations. All of the simulations were conducted on the EPA HPC system in a non-dedicated environment. On this HPC
system, there are 128 compute nodes (broadwell), organized in four MPI partitions of 32 nodes each with Intel E5-2697A v4
(16 cores, 2.6 GHz) and 32 compute nodes (cascadelake) with Intel 6248R (24 cores, 3.0 GHz) in one single MPI partition.
For consistency purposes, only broadwell nodes were used.

### 4.1   Model setup

Each MPAS and CMAQ model allows various combinatorial options for running the model. On the MPAS side, we followed
the same set of physics options that were used in a recent study (Gilliam et al., 2021): WSM6 single-moment microphysics
(Hong and Lim, 2006); Kain-Fritsch convection (Kain, 2004) modified to provide subgrid-scale cloud feedbacks to the radiation
schemes (Alapaty et al., 2012; Herwehe et al., 2014) and utilizing a scale-aware dynamic convective time scale (Bullock et al.,
2015); ACM2 planetary boundary layer (Pleim, 2007b); PX land surface model (Xiu and Pleim, 2001; Pleim and Xiu, 2003;
Pleim and Gilliam, 2009) coupled to a PX surface layer (Pleim, 2006) with NLCD40 (CONUS) and MODIS (rest of globe)
land use (Ran et al., 2016); and grid analysis nudging FDDA (Bullock Jr. et al., 2018). CMAQ is a chemical transport model
and it supports different chemical mechanisms where Carbon Bond is one of them. The Carbon Bond chemical mechanism has
long been used in CMAQ and we are using the CB6r5m version (Sarwar et al., 2015, 2019), which includes chlorine, bromine,
and iodine chemistry to better represent the atmosphere over marine environments, in this work. We also utilized other typical
settings (Appel et al., 2021) on the CMAQ side. Table 1 lists the key options which were used in this MPAS-CMAQ coupled
model.

   In general, CMAQ simulations take a much longer time than a meteorological model. One of the reasons is the stiffness of
the system of ordinary differential equations involved in the chemistry step and a large number of prognostic equations for
several chemical species. The MPAS-CMAQ coupled model provides a feature to allow users to choose the frequency of the





**Table 1.** Different options were used in the MPAS-CMAQ model.

| MPAS | CMAQ |
|---|---|
| microphycis scheme: WSM6 | chemical mechanism: CB6r5m |
| convection scheme: enhanced Kain-Fritsch | aerosol module: AER07 |
| land surface model scheme: Pleim-Xiu (PX) | deposition model: M3DRY |
| boundary layer scheme: ACM2 | chemistry solver: Rosenbrock |
| surface layer scheme: Pleim | |
| land use: NLCD40 and MODIS | |

data transfer between the MPAS and CMAQ models at run time. Users can balance model performance and model run time. In this study, we chose a frequency of 5:1, i.e., CMAQ will be called after every five MPAS time-steps.

The MPAS-CMAQ modeling requires emissions characterized on a global domain. Previous development of emissions for the hemispheric CMAQ utilized the global HTAP (Hemispheric Transport of Air Pollution, version 2) inventory (Janssens-
Maenhout et al., 2015) at a $0.1° × 0.1°$ grid resolution to provide anthropogenic emissions estimates outside of North America (Eyth et al., 2016). The HTAP inventories served as the basis for global anthropogenic emissions estimates outside of North America for MPAS-CMAQ. HTAP emissions were projected from 2010 to the model scenario year using scaling factors derived from the Community Emissions Data Systems historical emissions dataset (Hoesly et al., 2018). The SMOKE modeling environment was used to temporalize the HTAP emissions to hourly rates and speciate volatile organic compounds (VOC),
particulates (PM), and nitrogen oxides ($NO_x$). To prepare the gridded emissions for the MPAS mesh a spatial intersection was performed between the HTAP grid and MPAS mesh. Spatial allocation factors were calculated as the area of intersection between the grid and mesh cells divided by the total area of the HTAP grid cell. The spatial allocation factors were then applied to the HTAP emissions to calculate the total emissions rate at each MPAS cell. HTAP sectors related to aviation, biomass burning, energy generation, industrial processes, and waterborne navigation were vertically allocated to 44 layers using sector-
specific layer fractions. Recent inputs for MPAS-CMAQ replaced the HTAP emissions over China with emissions estimates from Tsinghua University (Zhao et al., 2018) provided on a 27 km × 27 km grid. The processes to temporalize, speciate, and spatially allocate the China emissions to the MPAS mesh were similar to the steps described for HTAP with replacement MPAS mesh allocation factors from the intersection of the China grid and the MPAS mesh. MPAS-CMAQ runs utilized inline MEGAN biogenic emissions rather than emissions generated in GEOS-Chem. The Fire Inventory from NCAR (FINN) v1.5
was used for wildland fire and agricultural burning emissions outside of the United States (Wiedinmyer et al., 2011). The FINN fire emissions inventories were processed through SMOKE to produce 2-D gridded emissions on the HTAP $0.1° × 0.1°$ grid. The gridded fire emissions were spatially and vertically allocated on the MPAS mesh using the area fractions derived for the HTAP grid and the biomass burning layer factors, respectively. Nitrogen oxide ($NO_x$) emissions from lightning strikes and their vertical allocation were estimated from a set of global monthly inventories (Price et al., 1997). These lightning $NO_x$ emissions
were apportioned to the $0.1° × 0.1°$ HTAP grid and mapped to the MPAS mesh using the spatial factors. Lightning $NO_x$ emissions were temporalized using seasonal lightning flash rates by continent and hour (Blakeslee et al., 2014). Speciation of





the lightning $NO_x$ used a 90/10 split to NO and $NO_2$. For all global sources the emissions were converted to a flux by dividing the rate by the area of the mesh cell.

Emissions over North America initially relied on the 2016 EPA Emissions Modeling Platform (EPA, 2021). Subsequent
North America emissions were developed from EPA's Air QUALity TimE Series Project (EQUATES) inventories and ancillary files (Foley et al., 2023). Speciation and temporal allocation methods were performed consistent with the emissions prepared for the regional modeling in the 2016 modeling platform and EQUATES. Spatial surrogates were generated using modeling platform weighting data and a shapefile of the target MPAS mesh. Area source sector emissions were spatially allocated in SMOKE to the mesh using the spatial surrogates. The SMOKE output gridded emissions were merged together across sectors
to produce a single area file with each row representing a mesh cell. A cross-reference of the area rows to the mesh cells was used to allocate emissions onto the MPAS mesh. As with the global emissions, the US and North American emissions were converted from a rate to a flux by dividing the emissions rate by the mesh cell area. Point emissions sources were processed through SMOKE for inline use in CMAQ without any additional post-processing.

### 4.2   Computational performance

We quantify the computational performance of the MPAS-CMAQ coupled system by measuring wall-clock time. For the 120 km uniform mesh, there are only 40962 mesh points, so we capture the wall-clock time for one-week simulations with MPAS alone and the MPAS-CMAQ coupled model (Fig. 4 left panel). For the 92-25 km variable mesh case, there are 163842 mesh points, which is about four times number of mesh points in the 120 km uniform case, so we did a one-day test instead for convenience (Fig. 4 right panel).

Overall, our results demonstrate the computational efficiency with a reasonable speedup as additional cores are being utilized. For the 120 km uniform mesh case, the strongest benefit of parallelization occurs when increasing from 64 cores to 128 cores for the coupled system. We observe smaller speed increases from larger numbers of cores when using stand-alone MPAS. The maximum number of cores tested here is 320 since efficiency is rapidly lost at this level of domain decomposition. For 320 cores, the MPAS simulation takes 43.3 minutes to complete and the MPAQ-CMAQ simulation takes 114.5 minutes to com-
plete. For the 92-25 km variable mesh case, stand-alone MPAS performance has reached a plateau with 256 cores and slightly increased the execution time with 512 cores. On the other hand, MPAS-CMAQ still performed with reasonable scalability even with 512 cores.

### 4.3   Preliminary Model Evaluation

The scope of this paper excludes a diagnostic accounting of the chemical processes in MPAS-CMAQ. Here we present prelim-
inary evaluation of the system's performance for surface air quality. We evaluate the MPAS-CMAQ system using two global configurations: a 120 km uniform mesh and a 92-25 km variable mesh with the finer area over North America. The 120 km uniform mesh allows for chemical spin up of the model at reduced computational cost. We run the 120 km configuration for a three-year period (2014-2016). The initial chemical state for the 120 km configuration is from a clean maritime profile that we





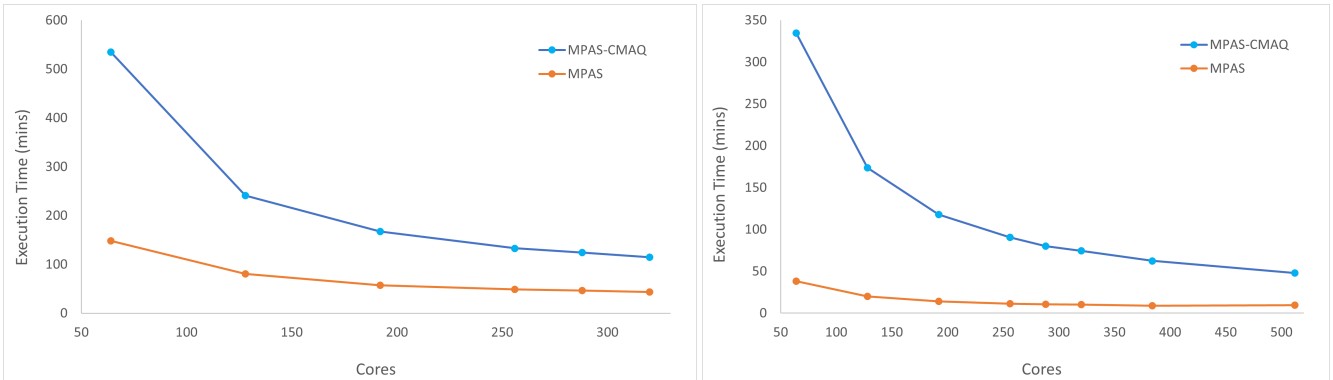

**Figure 4.** Computational performance of the MPAS (orange) and MPAS-CMAQ coupled model (blue) with 120 km uniform mesh (left) and 92-25 km variable mesh (right).

apply to every cell. Initial ozone concentrations are from the Copernicus Atmosphere Monitoring Service (CAMS) reanalysis
product (Inness et al., 2019) for January 1, 2014.

Modeled surface ozone closely follows the average interannual variability at all available Tropospheric Ozone Assessment Report, Phase II (TOAR2) sites (Fig. 5a). The time series show a consistent wintertime high bias of <10 ppb and a consistent low bias during spring and summer, resulting in an average high bias of 3.2 ppb for the entire period (Fig. 5b). We observe no drift in either the magnitude of the ozone or the bias during the three year simulation. The time tendencies in both measurements
and model are strikingly similar. Comparisons with ozonesondes indicate a low bias in free tropospheric ozone that peaks in springtime but the bias does not increase from 2014 to 2016 (not shown). These results give confidence in the stability of the global ozone budget in our modeling system.

The 120 km uniform mesh configuration of MPAS-CMAQ simulates summertime surface ozone that compares well with other contemporary global chemistry transport models and reanalysis products. Figure 6 shows average July surface ozone for
GEOS-CF (Keller et al., 2021), MPAS-CMAQ, CAMS Reanalysis, and MERRA-2 (Gelaro et al., 2017) reanalysis. MPAS-CMAQ generally simulates lower surface ozone concentrations than the CAMS and MERA-2 reanalysis products, but higher surface ozone than the GEOS-CF model.

The need for variable resolution within a consistent system motivates development of the MPAS-CMAQ framework. We test the 92-25 km variable resolution mesh with month-long simulations that we initialize from the chemical state of 120 km
spin-up runs. We simulate January 2016 and July 2016 to evaluate wintertime and summertime performance. Our area of interest for these tests is the contiguous United States (CONUS) region with the refined resolution. Figure 7a shows that the 92-25 km configuration of MPAS-CMAQ also has a high bias in wintertime surface ozone across the United States. This bias is consistent with the 120 km result (Fig. 5a) and the EQUATES dataset. We hypothesize that emission estimates and CMAQ model chemistry drive this bias rather than meteorological sensitivities that are unique to MPAS. Summertime surface ozone
in the 92-25 km configuration is less biased as compared to uniform mesh simulations except for the well-known California

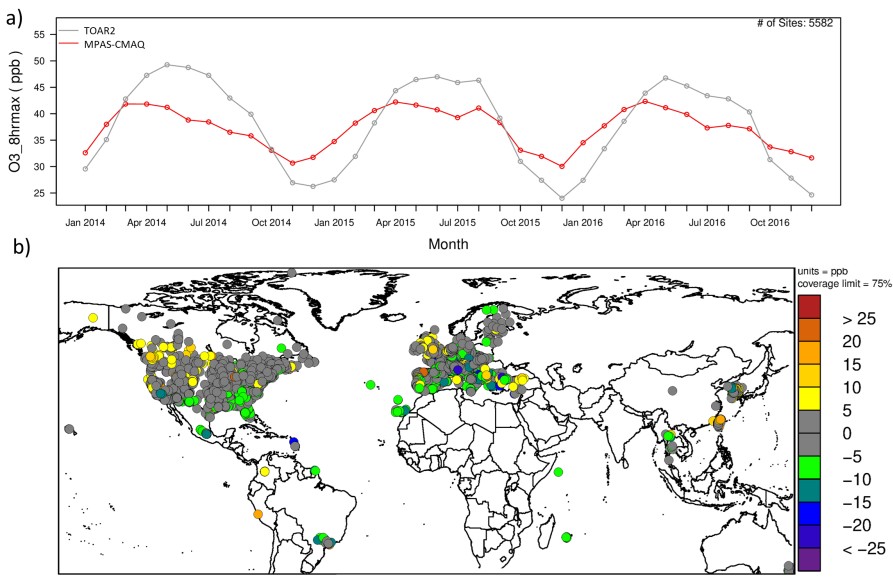

**Figure 5.** Modeled and observed surface ozone values for a) monthly average of all TOAR2 sites over time and b) average bias at each site for the three year period.

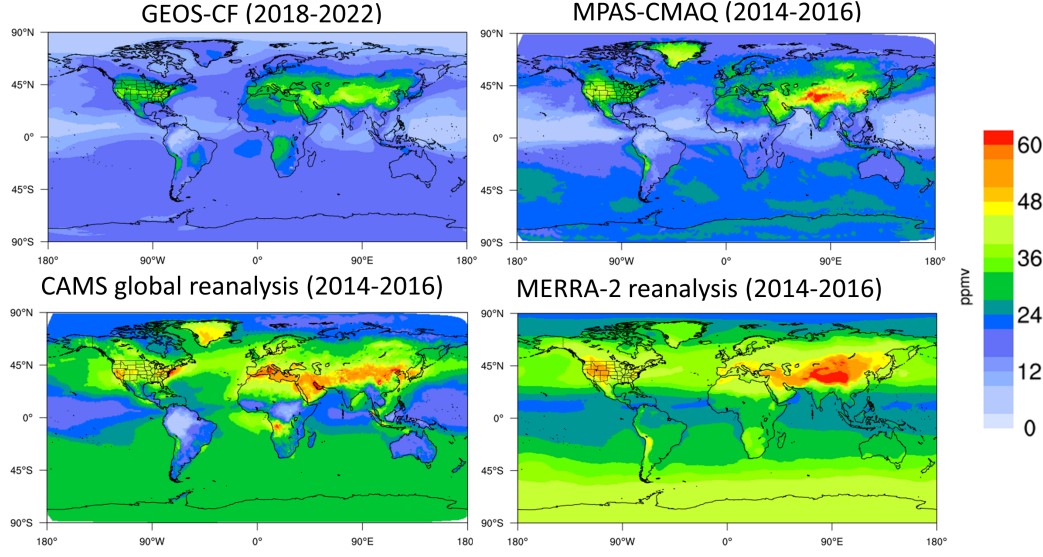

**Figure 6.** July average ozone for a) GEOS-CF, b) MPAS-CMAQ, c) CAMS reanalysis and d) MERRA-2 reanalysis.

central valley underestimate (Appel et al., 2021) (Fig. 7b). Again, we observe this bias in other CMAQ configurations and do not believe that it results from anything inherent to MPAS. The MPAS-CMAQ system reproduces the daily variability of



daily 8-hr max ozone reasonably well (Fig. 7c, d), with mean biases of 2.8 ppb and -0.8 ppb for January and July, respectively. These CONUS-average ozone biases are similar to previous evaluations of CMAQ, including versions 5.2.1 and 5.3.1 that had

maximum absolute biases of 4-6 ppb depending on the season and configuration options (Appel et al., 2021). The photolysis of particulate nitrate may improve model ozone underestimation in California and also the general springtime ozone under-estimation in the U.S. (Sarwar et al., 2024). We plan to include this chemistry in a future study as this study did not include it.

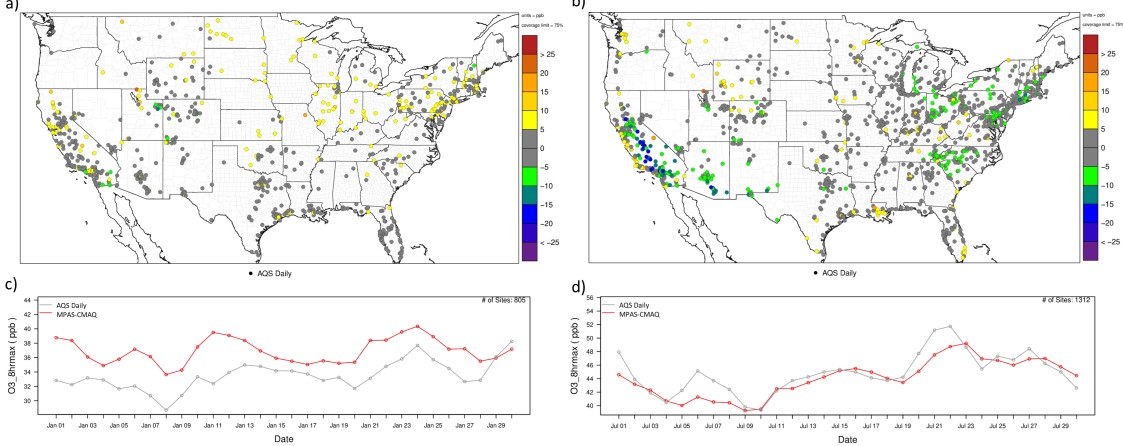

**Figure 7.** Average 8-hr max ozone bias at AQS sites for a) January 2016 and b) July 2016 on the 92-25 km variable resolution mesh. Corresponding average 8-hr max ozone mixing ratio for all AQS sites in c) January 2016 and d) July 2016.

The MPAS-CMAQ system must demonstrate skill in simulating PM$_{2.5}$ as well as ozone. The 92-25 km configuration pro-

duces a negative wintertime PM$_{2.5}$ bias in the western US and a general high bias east of the Rocky Mountains (Fig. 8a). Emissions estimates of residential wood combustion (RWC) were unavailable for this month and are excluded for the simulation. We expect that including these emissions would increase the PM over the CONUS. The 120 km uniform simulation that includes RWC performs similarly for PM$_{2.5}$ in January (not shown). In summertime, the bias is more consistently negative across the United States, with sporadic high biases dominating the average in the second half of the month (Fig. 8b). As with

ozone, the MPAS-CMAQ system reasonably reproduces the daily variability of daily average PM from the AQS network (Fig. 8c, d), with mean biases of -0.9 $\mu g/m^3$ and 0.1 $\mu g/m^3$ for January and July, respectively. The bias for our simulations compare well with Appel et al. (2021), who reported CONUS-average biases between 0.5-1.5 $\mu g/m^3$ for CMAQ versions 5.2.1 and 5.3.1 in the months of January and July.

## 5   Conclusion and Future Work

We have articulately demonstrated the construction of the MPAS-CMAQ modeling system and evaluated it for the ozone simulations. Since WRF-CMAQ modeling system was already evaluated and documented in the literature, this work only

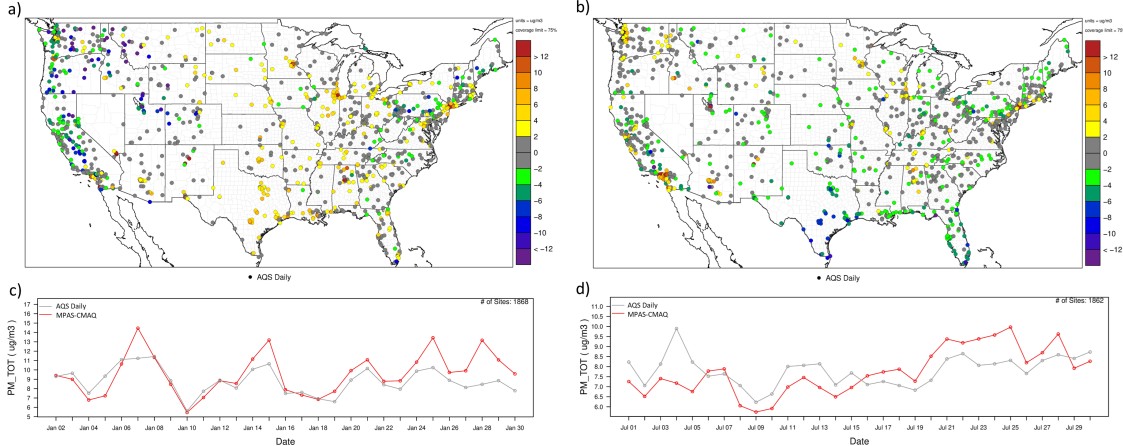

**Figure 8.** Average daily PM$_{2.5}$ bias at AQS sites for a) January 2016 and b) July 2016 on the 92-25 km variable resolution mesh. Corresponding average daily PM$_{2.5}$ concentration for all AQS sites in c) January 2016 and d) July 2016.

focused on the evaluation of MPAS-CAMQ modeling system. The newly developed AAQMS platform which includes the unified coupler, works for all types of CMAQ configurations. This MPAS-CMAQ coupled model has advantages over the WRF-CMAQ coupled model, including consistent transport algorithms, refinement without boundary interpolation and global
coverage without polar filters.

The preliminary results show the MPAS-CMAQ coupled model performed reasonably well with respect to ozone and PM$_{2.5}$ for North America, where the fine mesh is located, as well as the rest of the world. For future work, we will implement the aerosol radiative direct effect in this coupled model with a robust and flexible method to handle different CMAQ chemical mechanisms. We will also implement a switch so the aerosol radiative direct effect can be turned on or off at run time.

**Code and data availability**

- Entire MPAS-CMAQ with internal version CMAQ 5.4 is available at Zenodo (10.5281/zenodo.10982420).

- MIO is available at Zenodo (10.5281/zenodo.10994279).

- data which was used in generating Figure 4 - 9, is available at Zenodo (10.5281/zenodo.10994244).

**Author contributions** DCW defined the scope of the manuscript, developed the unified coupler, MIO, constructed the MPASC-
MAQ model. DCW, JW and JEP designed all the simulations, JW developed the MEGAN biogenic emission module and performed model evaluation. GP and JB processed emission. RB provided the FDDA updates. JH provided the EPA physics code. GS provided the marine chemistry CMAQ update. HF conducted initial scalar testing on MPAS. RG provided MPAS setup options. DK provided CMAQ lightning code update. DCW drafted the manuscript. JW, JEP, GP, RB, JH, CH, GS, HF,
RG, JB, and DK reviewed and edited the manuscript.



**Competing interests** The authors declare that they have no conflict of interest.

**Disclaimer** This paper has been subjected to an EPA review and approved for publication. The views expressed here are those of the authors and do not necessarily reflect the views and policies of the US Environmental Protection Agency (EPA).



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
