# Peer review of "Development of the MPAS-CMAQ Coupled System (V1.0) for Multiscale Global Air Quality Modeling"

_Geoscientific Model Development, 2024_

## Referee Comment (RC1)

**Referee Comment on egusphere-2024-52 "Development of the MPAS-CMAQ Coupled System (V1.0) for Multiscale Global Air Quality Modeling" by David C. Wong et al.**

The CMAQ model has a long history and is widely used across the world for research and regulatory purposes. The paper presents an important advancement of the CMAQ modelling system, going from limited area simulation to global simulation in a coupled setup with the meteorological model MPAS. A unified approach was developed, the Advanced Air Quality Modelling System (AAQMS), that enables the construction of the three different modeling systems: offline CMAQ, two-way coupled WRF-CMAQ, and global coupled MPAS-CMAQ.

The global coupled MPAS-CMAQ was evaluated using two global configurations, one with a uniform mesh for the globe and one with a variable mesh with finer resolution over North America. The study demonstrates good scalability and good performance for the uniform mesh simulations. The performance of the variable mesh case shows limitations with respect to surface ozone predictions in the United States, which have already been noted in simulations with earlier air quality modeling systems.

The manuscript is in general well written and logically organized. The layers of the AAQMS are clearly explained, with the exception of the coupler layer. The description of the unified coupler is not sufficient. It is hard to belief that the coupler does nothing else but data exchange in both directions. The claim that CMAQ inherits map projection, grid alignment, and grid spacing seamlessly in the coupled models needs to be substantiated.

The authors have chosen not to include WRF-CMAQ in the validation as this modeling system has been well documented and evaluated before. However, including WRF-CMAQ in the current evaluation over contiguous US would notably strengthen the credibility of MPAS-CMAQ and especially its performance in the variable mesh configuration. The revision of the manuscript should include a comparison of MPAS-CMAQ to the well-established WRF-CMAQ system for the limited area. In addition, my concerns listed below should be addressed.

Specific Comments:

1.) Section 2.1: It is compelling that MPAS and the CMAQ model are configured with the exact same grid configurations and coordinate systems. Maybe I missed updates in the coordinate system of CMAQ, but as far as I know, CMAQ uses Arakawa C horizontal staggering which previously required interpolation of the wind components to the locations of the CMAQ grid. Also, the coupler used for the WRF-CMAQ system uses the functionality embodied by the MCIP preprocesser that is used in the offline CMAQ. It might appear that CMAQ does not need the wind components since the transport is done in MPAS. However, sea spray parametrizations of

sea-salt particle emissions have a wind dependence and would still need wind components for their computation. There are also other additional variables (previously?) required in CMAQ such as the vertical coordinate Jacobian. Please add some more explanation on this.

2.) Figure 2 should display the variable mesh for the refinement over North America that was used in this study.

3.) P4, Line 78: Add a reference for "improve model performance for retrospective air quality applications."

4.) P4, Line 88: Define which processes are solved in the "CMAQ step". A table which lists the processes/solvers would be helpful. Are all the short-lived radicals also transported in MPAS?

5.) Section 3.1: The last paragraph (P6, Line 130-136) should be moved to the beginning of this section. The two major characteristic differences of the two coupled systems will require a different treatment by the coupler. It must be better explained how the characteristic differences in the two coupled systems lead to a different handling by the unified coupler. It is not clear to which extent the coupler supports software interoperability given that WRF-CMAQ and MPAS-CMAQ have different data structures (2-D vs 1-D) and different buffer demand for interpolation.

5.) Section 4.1: It is beneficial that users can balance model performance and model run time for the MPAS-CMAQ modeling. Have lower or higher frequency of the CMAQ call been tested and how is the sensitivity of the modeled ozone concentration field to changes of the frequency?

6.) Figure 5, part b): Show the average bias over the 3-year period separately for spring + summer and fall + winter. It seems that the negative and positive biases during the two periods cancelled out when averaging over the full year.

7.) Figure 6 and P10, Line 248-252: Give an explanation why high ozone over southern Africa in the biomass burning region does not occur in MPAS-CMAQ. Is biomass burning from this region missing in the model?

8.) Conclusion and Future Work: The authors claim that the MPAS-CMAQ coupled model has advantages over the WRF-CMAQ model. While this is correct from a theoretical point of view, the study did not demonstrate a superior model performance of MPAS-CMAQ compared to WRF-CMAQ. If the claim should be kept in the conclusion, then the validation of WRF-CMAQ runs for the CONUS region (Jan and Jul 2016) must be included.

Technical Corrections:

P2, Line 29-30: CMAQ and WRF should have capital letters in the sentence in brackets: "(note that cmaq …)".

P2, Line 36: Month of publication is usually not given in the citation.

---

## Author Comment (AC1)

Dear Referee #1, thank you so much for reviewing this paper and providing such a thorough and constructive review and comments. Here are our responses to your comments and suggestion point by point (in blue).

Referee Comment on egusphere-2024-52 "Development of the MPAS-CMAQ Coupled System (V1.0) for Multiscale Global Air Quality Modeling" by David C. Wong et al.

The CMAQ model has a long history and is widely used across the world for research and regulatory purposes. The paper presents an important advancement of the CMAQ modelling system, going from limited area simulation to global simulation in a coupled setup with the meteorological model MPAS. A unified approach was developed, the Advanced Air Quality Modelling System (AAQMS), that enables the construction of the three different modeling systems: offline CMAQ, two-way coupled WRF-CMAQ, and global coupled MPAS-CMAQ.

The global coupled MPAS-CMAQ was evaluated using two global configurations, one with a uniform mesh for the globe and one with a variable mesh with finer resolution over North America. The study demonstrates good scalability and good performance for the uniform mesh simulations. The performance of the variable mesh case shows limitations with respect to surface ozone predictions in the United States, which have already been noted in simulations with earlier air quality modeling systems.

The manuscript is in general well written and logically organized. The layers of the AAQMS are clearly explained, with the exception of the coupler layer. The description of the unified coupler is not sufficient. It is hard to belief that the coupler does nothing else but data exchange in both directions. The claim that CMAQ inherits map projection, grid alignment, and grid spacing seamlessly in the coupled models needs to be substantiated.

The authors have chosen not to include WRF-CMAQ in the validation as this modeling system has been well documented and evaluated before. However, including WRF-CMAQ in the current evaluation over contiguous US would notably strengthen the credibility of MPAS-CMAQ and especially its performance in the variable mesh configuration. The revision of the manuscript should include a comparison of MPAS-CMAQ to the well-established WRF-CMAQ system for the limited area. In addition, my concerns listed below should be addressed.

Specific Comments:

1.) Section 2.1: It is compelling that MPAS and the CMAQ model are configured with the exact same grid configurations and coordinate systems. Maybe I missed updates in the coordinate system of CMAQ, but as far as I know, CMAQ uses Arakawa C horizontal staggering which previously required interpolation of the wind components to the locations of the CMAQ grid. Also, the coupler used for the WRF-CMAQ system uses the functionality embodied by the MCIP preprocesser that is used in the offline CMAQ. It might appear that CMAQ does not need the wind components since the transport is done in MPAS. However, sea spray parametrizations of sea-salt particle emissions have a wind dependence and would still need wind components for their computation. There are also other additional variables (previously?) required in CMAQ such as the vertical coordinate Jacobian. Please add some more explanation on this.

It is correct that both the offline CMAQ and WRF-CMAQ coupled model use Arakawa C horizontal staggering grid structure. However, in the MPAS-CMAQ coupled model, all grid-scale transport processes are performed by MPAS and these transport processes are disabled in CMAQ. This makes CMAQ act like a chemistry "box" model and behaves as a subroutine in the MPAS structure. Therefore, the MPAS and CMAQ models are configured with the exact same grid configuration. Only select meteorological fields such as wind speed and temperature are transferred from MPAS to CMAQ for its calculation of processes such as inline emissions, plume rise, wet and dry deposition, and gas-particle partitioning. Regarding the Jacobian, it is set to 1 since WRF's sigma coordinate system is not used.

2.) Figure 2 should display the variable mesh for the refinement over North America that was used in this study.

We agree that the ideal figure would show the meshes used for the experiments. Unfortunately, the actual meshes are not ideal for demonstrating the mesh refinement because the cells are small relative to the cartoon. For example, here are the 120 km uniform (left) and 92-25 km (right) meshes used in this study:

[Figure]

[Figure]

These are quite busy, so we chose the representative cartoons instead, with refinement over Asia makes it obvious that these are not the meshes we used.

3.) P4, Line 78: Add a reference for "improve model performance for retrospective air quality applications."

The model configuration we use here has evolved over time at EPA and there is not a single reference demonstrating the combined value of these physics options versus other permutations of options. It is important to clarify that these physics implementations were designed specifically for air quality modeling and with the intention of improving the simulation of meteorological conditions that are

important for atmospheric chemistry. Meteorological physics options are often designed for skillful representation of conditions that are associated with extreme weather (convective boundary layers, thunderstorms, etc.). Simulating atmospheric chemistry is much more dependent on representation of, for example, the stable nocturnal boundary layer and chemical transport within it. The paper describing our boundary layer model ACM2, states:

> *The particular advantage of the ACM2 over eddy diffusion schemes with a counter-gradient adjustment term (e.g., Holtslag and Boville 1993; Troen and Mahrt 1986; Noh et al. 2003) is its applicability to any quantity, either meteorological or chemical. While the latter type of model has been very successful for meteorological modeling, it is not clear how it can be extended to atmospheric chemistry modeling because the counter-gradient term is directly related to the surface flux of the modeled variable. This makes sense for heat or moisture, which involve turbulent surface fluxes, but not for chemical species where surface sources are often pollutant emissions that are not related to turbulent fluxes. The ACM2, on the other hand, is a mass flux scheme that does not depend on the surface sources or sinks and can easily accommodate any source/sink profile, including elevated sources.*

We have reworded to make this distinction clearer:

> *Existing MPAS physics options are generally expected to be skillful at simulating atmospheric conditions associated with extreme weather. The physics options used here were designed to improve meteorological conditions that influence atmospheric chemistry and EPA commonly uses these options for retrospective air quality modeling (e.g. Appel et al., 2021).*

4.) P4, Line 88: Define which processes are solved in the "CMAQ step". A table which lists the processes/solvers would be helpful. Are all the short-lived radicals also transported in MPAS?

The phrase "the transport portion of the code within CMAQ is turned off" has been revised as "the transport portion of the code within CMAQ (horizontal and vertical advection as well as horizontal diffusion) is turned off" to clarify exactly what these transport processes are. All the chemical species defined in CMAQ including the short-lived radicals, are being transported in MPAS.

In traditional CMAQ the fast-reacting radical species are not transported, with the assumption that the characteristic time for fast reacting species is significantly smaller than that of transport processes. In the future we plan to evaluate thoroughly if we disable the transport of these short-lived species in MPAS-CMAQ for consistency with traditional CMAQ behavior.

5.) Section 3.1: The last paragraph (P6, Line 130-136) should be moved to the beginning of this section. The two major characteristic differences of the two coupled systems will require a different treatment by the coupler. It must be better explained how the characteristic differences in the two coupled systems lead to a different handling by the unified coupler. It is not clear to which extent the coupler supports software interoperability given that WRF-CMAQ and MPAS-CMAQ have different data structures (2-D vs 1-D) and different buffer demand for interpolation.

We respectfully disagree with the reviewer's assessment that the last paragraph (P6, Line 130-136) should be moved to the beginning of this section. The reason is the current layout provides a nice transition from what is out there and what we used to where we are heading. Specifically, first it gives readers a sense of what is available out there and why we did not choose them. It also described the coupler construct in the WRF-CMAQ coupled model and the benefits of a unified coupler with respect to software interoperability. At last, the details of the specific constructs of the unified coupler are given. We revised the last paragraph with additional clarification.

"The unified coupler adopted simple and straightforward strategies to handle these two major characteristic differences. For the first characteristic, in both models, 2D arrays are used. For MPAS, which arranges all mesh points linearly as a 1D array, the size of the second dimension is set to 1. For the second characteristic, we put the circular buffer dimension as the last dimension in an array with 0:1 declaration to facilitate time interpolation in the WRF-CMAQ coupled model, and defined as 1 for the MPAS-CMAQ coupled model to indicate no interpolation is necessary. With the MPAS-CMAQ coupled model, the transport is handled in MPAS and the time step in CMAQ is fully synchronized with MPAS."

5.) Section 4.1: It is beneficial that users can balance model performance and model run time for the MPAS-CMAQ modeling. Have lower or higher frequency of the CMAQ call been tested and how is the sensitivity of the modeled ozone concentration field to changes of the frequency?

Thank you for this comment, as it helped identify an error in the text. We have clarified that the coupling frequency is 3:1 for the 92-25 km mesh but 1:1 in the uniform 120 km mesh.

In striving for efficient 92-25 km simulations, we tested various coupling frequencies, including calling CMAQ every half hour. We consider this to be an upper limit for a reasonable CMAQ time step at this resolution. The systematic biases shown here for ozone and PM2.5 were consistent and independent of coupling frequency, so we settled for a 7.5 minute CMAQ timestep. For reference, EPA typically runs the 12 km domain with a 5 minute CMAQ time step, and the 108 km hemispheric domain with a 15 minute time step.

We may find larger sensitivities and computational tradeoffs at higher resolutions. Later this year we plan to carry out more systematic testing of this sensitivity for domains that include near-convection-resolving horizontal resolutions (~4 km).

Corrected text:

> In this study, we chose a frequency of 3:1 for the variable resolution mesh, i.e., CMAQ will be called after every three MPAS time-steps. For the uniform mesh we use a coupling frequency of 1:1. Varying this frequency does not result in systematic changes to the performance metrics presented here, but should be investigated more thoroughly in a future study.

6.) Figure 5, part b): Show the average bias over the 3-year period separately for spring + summer and fall + winter. It seems that the negative and positive biases during the two periods cancelled out when averaging over the full year.

We agree that the averaging of winter and summer biases results in an uninformative spatial plot. We have removed the full-year average panel and added summer and winter panels (Fig. 5b,c). These panels help highlight that the wintertime biases occur at the surface throughout the domain, while the summertime low biases are dominated by areas outside of the United States. The text discussing Fig. 5 has been updated to reflect this new information:

"Figure 5b shows that the high surface ozone bias in winter is consistent across the Northern Hemisphere, while Fig. 5c shows that the summertime performance is characterized by large negative biases over Europe."

7.) Figure 6 and P10, Line 248-252: Give an explanation why high ozone over southern Africa in the biomass burning region does not occur in MPAS-CMAQ. Is biomass burning from this region missing in the model?

Biomass burning (wildland and agricultural fires) emissions from FINN v1.5 included this region of Africa (see attached plot of NO flux from the g_ptfire sector on the 60x12 mesh for 12 June 2016). However, FINN v1.5 is known to underpredict biomass burning emissions in sub-Saharan Africa relative to other global inventories (see Fig 4. here: https://doi.org/10.5194/gmd-16-3873-2023 and section 3.1 here: https://doi.org/10.5194/gmd-4-625-2011). Additionally, biomass burning emissions from the Democratic Republic of Congo were inadvertently omitted in the initial release of the 2016 FINN v1.5 per 2017 communication between Jeff Vukovich at EPA-OAQPS and Christine Wiedenmeyer at UCAR. This can be confirmed with the attached image where the outline of DRC is visible as an abrupt transition of NO flux rates. If the lower than expected O3 in this region is related to biomass burning then this can be partially explained with the emissions underprediction in the now deprecated FINN v1.5.

[Figure]

8.) Conclusion and Future Work: The authors claim that the MPAS-CMAQ coupled model has advantages over the WRF-CMAQ model. While this is correct from a theoretical point of view, the study did not demonstrate a superior model performance of MPAS-CMAQ compared to WRF-CMAQ. If the claim should be kept in the conclusion, then the validation of WRF-CMAQ runs for the CONUS region (Jan and Jul 2016) must be included.

We agree that it is important to compare MPAS-CMAQ performance with more traditional implementations of CMAQ that use WRF meteorology. However, we currently lack inputs (emissions and boundary conditions) that are adequately similar to support a quantitative comparison. We often see that differences in emission and boundary condition inputs affect statistical performance more strongly than grid configuration. We therefore feel that a rigorous evaluation between MPAS-CMAQ and WRF-CMAQ merits a separate future publication when more comparable input datasets become available.

To provide a qualitative comparison of the MPAS-CMAQ results presented in this manuscript with results from a contemporary application of WRF-CMAQ, below we have reproduced our Figs. 7 and 8 using data from the EPA's Air QUAlity TimE Series (EQUATES) Project (https://www.epa.gov/cmaq/equates). These

simulations were carried out with CMAQ version 5.3.2 on a 12 km resolution WRF grid. For comparison with Figure 7 (ozone):

[Figure]

We see that the wintertime bias is higher in MPAS-CMAQ, particularly in the northern parts of the domain. The summertime biases are improved in MPAS-CMAQ, with reduction of the high ozone bias observed in the eastern United States. The strong negative biases in California during summertime are consistent regardless of meteorological forcing.

Below is an evaluation of EQUATES PM 2.5 performance, for comparison with our Figure 8:

[Figure]

Here we see very similar patterns for both winter and summer. MPAS-CMAQ and WRF-CMAQ show strong wintertime negative biases along the west coast and have a tendency for broad regions of negative bias in the summertime. The magnitudes of bias are similar for both configurations. This is reassuring when considering the numerous differences between the configurations, including horizontal resolution, vertical resolution, treatment of stratospheric ozone, global emissions inventories, grid structure, biogenic emission inputs, and CMAQ science updates.

We feel that a rigorous evaluation between MPAS-CMAQ and WRF-CMAQ merits a separate publication. We have modified the wording in the conclusions to accurately reflect the state of our MPAS-CMAQ and WRF-CMAQ comparisons (changes in bold):

> This MPAS-CMAQ coupled model has **theoretical** advantages over the WRF-CMAQ coupled model, including consistent transport algorithms, refinement without boundary interpolation and global coverage without polar filters. **It remains to be shown whether these numerical advantages will translate to improved statistical performance metrics for simulations of retrospective air quality.**

Technical Corrections:

P2, Line 29-30: CMAQ and WRF should have capital letters in the sentence in brackets: "(note that cmaq …)".

Done.

P2, Line 36: Month of publication is usually not given in the citation.

Done.

---

## Author Comment (AC2)

Dear Referee #2, thank you so much for reviewing this paper and providing such a thorough and constructive review and comments. Here are our responses to your comments and suggestions point by point (in blue).

This manuscript describes a new coupled model system based on the MPAS meteorological model and CMAQ air quality model. The coupled system with a variable mesh grid has numerous advantages over other model systems such as a consistent transport scheme for pollutants and meteorological variables and the advantages of finer resolution without the need to nest domains, which can introduce interpolation errors. The manuscript is very well written. I recommend publication with only minor changes.

Line 40. I feel like there should be a reference to MPAS somewhere in this paragraph.

The text has been modified as:

"The National Center for Atmospheric Research (NCAR) has recently developed a new global meteorological model, the Model for Prediction Across Scales – Atmosphere (Skamarock et al., 2012) (MPAS-A, hereafter referred to as MPAS),"

and the following new reference is added:

Skamarock, W. C., Klemp, J. B., Duda, M. G., Fowler, L. D., Park, S.-H., and Ringler, T. D.: A Multiscale Nonhydrostatic Atmospheric Model Using Centroidal Voronoi Tesselations and C-Grid Staggering, Monthly Weather Review, 140, 3090–3105, https://doi.org/https://doi.org/10.1175/MWR-D-11-00215.1, 2012.

Line 75. Can you please check this sentence. The sentence states enhancements for physics options and then states "namely the Pleim-Xiu land-surface model" which is a surface model and not physics model.

The physics options were referring to the physics options in the namelist. To avoid misunderstanding, the phrase "physics options" has been revised as "physics models which are associated with specific physics options in the namelist," in the text.

Line 125. Can you provide a bit more detail. Are the soluble and insoluble species in the PM phase? What are the other components needed in MPAS-CMAQ system to model the online aerosol direct effect?

This soluble and insoluble species classification is determined automatically with respect to aerosol species which depends on the version of aerosol scheme used in CMAQ. The other three categories are sea salt, EC, and aerosol water. These five categories are the same as in the WRF-CMAQ coupled model.

Line 185.  One of the advantages of the MPAS-CMAQ system is the variable mesh which enables finer scale without the need for nesting grids. This removes the need for interpolation of lateral boundary conditions. However, for the tests performed here, the emissions were interpolated from rectangular grids to the MPAS grid which introduces errors and may counter the gain from not needing nesting. In future work, it would be beneficial to do the emissions processing directly onto the MPAS grid rather than interpolate from other grids.

Global emissions outside of North America rely on HTAP data (https://doi.org/10.5194/acp-15-11411-2015), which is provided on a 0.1 x 0.1-degree grid. To our knowledge there is no consistent source of global emissions data available that could be spatially apportioned to a mesh without interpolation. The 0.1 degree by 0.1-degree grid spacing is generally more of an aggregation of emission estimates rather than interpolation since most of the mesh cells are larger than 0.1 degree by 0.1 degree so there are not many mesh cells that get redistributed in a global MPAS mesh. Within North America regional inventories are directly apportioned to the MPAS mesh using spatial surrogates. This method is limited by both the resolution of the weighting data used in the surrogates and the need to generate new surrogates for every mesh configuration. Also a large number of emission sources in North America are point sources which are mapped directly to the MPAS mesh cells without interpolation.

Line 218.  I feel there should be more information on how plume rise is calculated in MPAS-CMAQ since this is a key process where meteorology impacts pollutant dispersion. Is plume rise handled in MPAS or CMAQ? If in CMAQ, what parameters are transferred to CMAQ to simulate the vertical mixing. Does the water vapor in the point source feedback to the meteorology and provide more latent heat?

The plume rise calculation is based on Brigg's algorithm and is performed on the CMAQ side (in the CMAQ offline model as well as the WRF-CMAQ coupled model), taking into account temperature and wind profiles transferred from MPAS and stack information (stack height, diameter, temperature, flow rate, and exit velocity) provided for each point source being modeled. Information on water vapor associated with point source plumes is not available from point source emission inventories and is therefore not part of the information being fed back to the meteorological model.

Line 246.  Does the MPAS-CMAQ system include ozone data assimilation in the stratosphere. This may improve the ozone low bias in winter/spring, particularly in free troposphere.

We thank the reviewer for identifying this omission. The new text includes a description of our data assimilation approach in the stratosphere in the Section 4.1:

"CMAQ does not include a full representation of stratospheric chemistry, and the potential vorticity scaling approach that is used in hemispheric-scale CMAQ domains to estimate ozone mixing ratios in the

upper layers (Xing et al., 2016) is not viable over the equator. In MPAS-CMAQ we ingest time-dependent values of stratospheric ozone from the Copernicus Atmosphere Monitoring Service (CAMS) reanalysis product (Inness et al., 2019) where model pressure is less than 300 hPa and model ozone is above 200 ppb."

We experimented with many versions of this approach, including using other ozone products, more traditional definitions of the tropopause height, and FDDA. For better or worse, the bias in the free troposphere was relatively insensitive to the approach taken.

I would recommend larger size for some of the figures (e.g. Figure 7,8), maybe not side by side.

We have increased the size of the figures to provide a clearer view.

Grammar Corrections

Line 29. Change "a" to "an".

Can't find this item. We believe the reviewer asked to change "a Eulerian" to "an Eulerian" and it is done

Line 50. Change "entities" to "objectives"

Done.

Line 230.  Remove "has".

Done.

Line 251.  Correct MERA to MERRA

Done.

Line 282. Change CAMQ to CMAQ

Done.